# π-Conjugated Polymer Nanoparticles from Design, Synthesis to Biomedical Applications: Sensing, Imaging, and Therapy

**DOI:** 10.3390/microorganisms11082006

**Published:** 2023-08-03

**Authors:** Nada Elgiddawy, Noha Elnagar, Hafsa Korri-Youssoufi, Abderrahim Yassar

**Affiliations:** 1CNRS, Institut de Chimie Moléculaire et des Matériaux d’Orsay (ICMMO), Université Paris-Saclay, ECBB, 91400 Orsay, France; 2Department of Biotechnology and Life Sciences, Faculty of Postgraduate Studies for Advanced Sciences (PSAS), Beni-Suef University, Beni-Suef 62 511, Egypt; 3Materials Science and Nanotechnology Department, Faculty of Postgraduate Studies for Advanced Sciences (PSAS), Beni-Suef University, Beni-Suef 62 511, Egypt; 4LPICM, CNRS, Ecole Polytechnique, Institut Polytechnique de Paris, Route de Saclay, 91128 Palaiseau, France; abderrahim.yassar@polytechnique.edu

**Keywords:** conjugated polymer nanoparticles, biomedical application, organic semiconducting, fluorescence sensing, electrochemical sensing, drug delivery, bioimaging

## Abstract

In the past decade, π-conjugated polymer nanoparticles (CPNs) have been considered as promising nanomaterials for biomedical applications, and are widely used as probe materials for bioimaging and drug delivery. Due to their distinctive photophysical and physicochemical characteristics, good compatibility, and ease of functionalization, CPNs are gaining popularity and being used in more and more cutting-edge biomedical sectors. Common synthetic techniques can be used to synthesize CPNs with adjustable particle size and dispersion. More importantly, the recent development of CPNs for sensing and imaging applications has rendered them as a promising device for use in healthcare. This review provides a synopsis of the preparation and functionalization of CPNs and summarizes the recent advancements of CPNs for biomedical applications. In particular, we discuss their major role in bioimaging, therapeutics, fluorescence, and electrochemical sensing. As a conclusion, we highlight the challenges and future perspectives of biomedical applications of CPNs.

## 1. Introduction

The last decade has witnessed great progress in the use of conjugated polymer nanoparticles (CPNs), sometimes referred to in the literature as semiconducting polymer nanoparticles or polymer dots (Pdots), as an emerging class of multifunctional nanoscale materials for biomedical applications, including fluorescence and photoacoustic imaging, biosensors, and photothermal therapy for microbial infections and cancer [1,2,3,4]. CPNs exhibit outstanding properties combined with a simple preparation method, easy surface modification, and water dispersibility [5]. Their straightforward synthesis [4] in desired sizes and tunable properties makes them highly attractive materials for the aforementioned applications [6,7]. Their photophysical properties can be tailor-made by an appropriate rational molecular design of the structure of the conjugated polymers (CPs) or aggregate states. Until now, CPNs and, more generally, soft fluorescent nanoparticles (NPs) have emerged as novel biomedical analysis tools due to their exceptional photophysical properties, outstanding photochemical stability, extraordinary fluorescence efficiency, tunable spectral properties, and amplified fluorescent quenching effect with less toxicity. They have been exploited in fluorescence imaging [8], cancer cell bioimaging [9], photodynamics, photothermal therapy [10], antimicrobial therapy, and bacteria killing, as shown in Figure 1 [11]. For biomedical applications, CPNs present many advantages over small organic molecule nanoparticles. They can be obtained with accessible synthetic schemes, thus providing a much larger platform for material manipulation to meet the specific requirements for different biomedical applications. Their chemical structures, band gap energy, functionalities, topology, and morphology can easily be tuned compared to those of the organic dye nanoparticles. Furthermore, many recent reports have shown that the effectiveness of CPNs as fluorescent probes for bioimaging can surpass that of small molecules and inorganic quantum dots due to their extraordinary fluorescence brightness, excellent photochemical stability, high-fluorescence quantum yield, biocompatibility, and low toxicity to cells and tissues [12]. Among the key factors that affect their utility as biomedical tools are fluorescence emission wavelength, the brightness of the fluorescence, the size of the particle, and the Stokes shift, which should be small to capture the pertinent fluorescent signals and avoid cross-talk between the excitation light and fluorescence. When CPNs are designed as fluorescent probes, they should be as small as possible for complete elimination from the body by renal filtration, i.e., their size should be less than about 5.5 nm.

According to Web of Science (accessed spring 2023), the number of articles published in the field of NPs and “conjugated polymers” has exponentially increased, indicating the importance and interest in such emerging fields, as shown in Figure 2. The interest in nanoparticle synthesis began in 2002, with the work of Landfester et al. on the mini-emulsion technique. Since this seminal work, a number of reviews dealing with the developments and applications of organic fluorescent nanoparticles have been published. These reviews have highlighted small organic molecules [9,14,15], conjugated polymer nanoparticles [3,16], aggregation-induced emission dots [17,18,19], carbon dots [20,21], and up-conversion nanoparticles [22]. Some of these reviews cover very broad topics, including the synthesis of organic and polymeric materials and the preparation of NPs, whereas others focus on applications, including sensing [23], imaging [24], and diagnostic and therapeutic potential (phototherapy). In contrast, this review showcases the recent achievements in the development of high-performance tailor-made CPs with unique optical properties, combined with their incorporation into NPs and use in biomedical applications. A particular focus is placed on material design and synthesis, structures, and topologies, as well as the functionalities and biomedical applications of CPs.

## 2. Preparation Methods of CPNs

Two main strategies are used to produce CPNs: direct polymerization to NPs and the dispersion of preformed polymers (namely post-polymerization). Each strategy has its unique advantages and limitations. Direct polymerization does not require the use of polymers that are only soluble in organic solvents. The post-polymerization approach, however, produces NP dispersion from existing polymers using mini-emulsification, nano-precipitation, self-assembly processes, or vesicle and micelle formation methods. Therefore, this approach does not require any special equipment or expertise in organic and polymer synthesis, and it can rely on commercially available polymers. 

### 2.1. Direct Synthesis of Conjugated Polymer NPs 

In the direct synthesis of CPNs approach, the reaction of the polymerization and the NPs formation occurred simultaneously in a disperse heterophase system. In such approaches, the polymerization occurs inside the mini- or micro-emulsion formed in continuous immiscible phases. Mini- and micro-emulsion polymerization differ mainly in their stability, while the microemulsion is thermodynamically stable and forms spontaneous microdroplets under appropriate conditions. However, mini-emulsion is a metastable phase which can be achieved with the help of shearing the immiscible phase through the sonication or the homogenization of the solution. While micro-emulsion polymerization produces NPs with a diameter as small as 5 nm, these sizes are typically greater than 50 nm in mini-emulsion polymerization. Finally, both emulsion polymerizations are not only useful techniques for preparing NPs from existing polymers, but they can also serve as a template reaction medium for synthetizing new CPNs from novel monomers [2].

While the majority of the literature described the creation of NPs utilizing oil-in-water method, Mullen et al. showed that a non-aqueous emulsion (oil-in-oil) can be used to elaborate conjugated polymer NPs through direct polymerization of monomers [25]. In their work, cyclohexane and acetonitrile were used as a continuous and dispersed phase, respectively, while polyisoprene-block-poly(methyl methacrylate) (PI-*b*-PMMA) was utilized as an emulsifying agent. NPs of Poly(3,4-ethylenedioxythiophene) (PEDOT), polyacetylene and poly(thiophene-3-acetic acid) were prepared by catalytic and oxidative polymerization [26]. Additionally, the emulsifying agent (PI-*b*-PMMA) was washed with a THF solution leading to CPNs with an average diameter of 23 nm. In 2006, Huber et al. synthesized processable submicron polyacetylene NPs using catalytic emulsion polymerization [27]. The key to the success of their strategy relies on the use of catalytic complexes that are less oxophilic and very reactive towards the polymerization of the acetylene monomers. For this purpose, a small volume of an emulsion of hexane/ethanol was prepared; then, a catalytic amount of Pd^II^, 1,3-bis(di-tert-butyl)phosphino-propane and Pd(OAc)_2_ was added followed by the addition of an aqueous solution of surfactant, sodium dodecyl sulfate (SDS), and methane sulfonic acid. The mixture was subsequently sonicated to produce a mini-emulsion. Polymerization of acetylene under ambient pressure resulted in the formation of an intensely colored, black CPN dispersion. The diameter of the NPs was estimated to be 20 nm by using the transmission electron microscopy (TEM) [27]. The same group extended this concept to the aqueous step-growth polymerization of conjugated polymer NPs [28]. They demonstrated that the highly fluorescent CPNs could be prepared by direct polymerization in aqueous mini-emulsion, under Glaser coupling conditions as a suitable step-growth reaction. The molecular weights estimated by gel permeation chromatography are in the range of 10^4^ to 10^5^ g mol^−1^, and NP sizes determined by TEM are around 30 nm.

### 2.2. Conventional Synthesis Method of CNPs 

Conventional methods to obtain CNPs involve post-polymerization to produce CPNs from previously synthesized CPs. They can be generated using mini- or macro-emulsion, nanoprecipitation, or self-assembly processes.

#### 2.2.1. Mini-Emulsion Method

The mini-emulsion approach is the oldest known and most widely used method to prepare CPNs. In their seminal work, Landfester and coworkers demonstrated the preparation of CPNs using the mini-emulsion technique for the first time. They reported the preparation of a series of water-dispersible CPs, methyl-substituted ladder-type poly(para-phenylene), polyfluorene, and polycyclopentadithiophene, with NPs sizes in the range of 75–250 nm by controlling the amount of the surfactant [29]. In such a method, the CP is solubilized in a good organic solvent, which is then injected in the water phase containing a surfactant, a bad solvent of CPs. High shear forces via ultra-sonication were then applied to obtain homogeneous and monodisperse droplets stabilized by surfactant molecules. In this way, a suspension of water dispersible NPs is formed by a slow evaporation of organic solvent. The polymer/surfactant ratio alters the size of the NPs, which typically range from 20 nm to several hundred nanometers. The mini-emulsion polymerization method has two benefits over nano-precipitation: the large choice of water immiscible organic solvents and the use of a surfactant to prevent the coalescence of emulsion droplets. The major drawback related to this method is destabilization risks due to Ostwald ripening and droplet coalescence [30]. The risk of flocculation may be avoided by an appropriate choice of surfactants used, while Ostwald ripening can be suppressed by the addition of a hydrophobic agent to the dispersed phase. The hydrophobic additive promotes the formation of an osmotic pressure within the droplets, which balances the Laplace pressure, the difference in pressure between a droplet’s interior and exterior, and prevents diffusion from one droplet to the surrounding watery medium.

Recently, a cradle-to-grave study of CPN-based indaceno-dithiophene and diketopyrrolopyrrole copolymers was reported by Turner’s group [31]. The study aims to achieve high-performance organic semiconductor (OSC) device based on CPNs and therefore, confirms the effect of chemical synthesis on their physico-chemical properties such as light absorption and emission. 

The process utilizes a water-in-oil mini-emulsion Suzuki–Miyaura polymerization, fewer steps, lower reaction temperatures, a significant reduction in volatile organic compounds (VOCs) (>99%) and avoids halogenated solvents; Figure 3.

Suzuki–Miyaura polymerization of borylated benzothiadiazole (BT(Bpin)_2_) and di-halogenated indacenodithiophene or diketopyrrolopyrrole-thiophene, under conventional conditions, in a water-in oil mini-emulsion, was performed to prepare a series of copolymers; Figure 4. The optimized mini-emulsion polymerization conditions led to a complete monomer consumption and the formation of copolymers with molecular weights comparable to those obtained by conventional polymerization methods. This process generates a highly concentrated NP dispersion; these NPs are surfactant coated on their surface. These dispersions were used to fabricate thin films for electronic devices. The material performance was found to be similar to that obtained using large amount of VOCs and halogenated solvents. The complete approach developed addresses all environmental concerns and enables a viable guideline for the delivery of future CP materials using only aqueous media for synthesis, purification, and thin-film processing.

#### 2.2.2. Nano-Precipitation Methods: Solvent Mediated Self-Assembly

The interaction between the CP chains and solvent molecules is a crucial parameter in the formation of NPs. In the nano-precipitation approach, a given CP is dissolved in a good organic solvent and then injected into a bad CP solvent. By contrast, this bad solvent is miscible with the solvent used to dissolve the CP, leading to fine precipitation of CPNs. To assist the formation of the NPs, the resulting mixture is vigorously sonicated. The hydrophobic effect, which is caused by the tendency of CP chains in solution to avoid contact with solvent molecules, is the main driving force for the formation of NPs. In fact, the formation of the NPs is attributed to the aggregation of the CP chains caused by the significant change in solvent polarity, and, consequently, to avoid their exposure to solvent molecules, they fold into spherical shapes. This method does not involve the use of any extra additives such as surfactants and hydrophobic agents and can be applied to a wide range of organic solvent-soluble CPs. By this method, it is possible to fine-tune the size of NPs by adjusting the polymer concentration and an appropriate choice of the molecular weight. Using this approach, one can easily control the NP size down to 40 nm by changing the polymer concentration. Furthermore, many studies indicated that the initial concentration of CP affects the size of the NPs [32,33]. The diameter of NPs is greater the higher the initial CP concentration.

#### 2.2.3. Self-Assembly Method through Supramolecular Interaction

Comparatively, the self-assembly strategy, where the polymers assemble into predefined structures guided by specific supramolecular interactions, has been less employed. In this strategy, oppositely charged π-conjugated polyelectrolytes and co-assembling reagents are independently dissolved in a solvent (Figure 5). To create charge-neutral NPs, the resulting solutions or dispersions are successively mixed in a specific ratio while being stirred. The functionalized CPNs can be easily separated by centrifugation. Using this method, Chong and coworkers synthetized photo-activated CPNs with antitumor activity. In their process, an aqueous solution of disodium salt, 3,3′-dithiodipropionic acid (SDPA) was added under vigorous stirring to a solution of a copolymer-containing fluorene and boron-dipyrromethene repeat units in the backbone (PBF). CPNs with a diameter of 100 nm were obtained by centrifugation [34].

### 2.3. Vesicles and Micelles Formation

Alternatively, rod–rod conjugated block copolymers with a hydrophobic and a hydrophilic block have the ability to self-assemble into block copolymer micelles with a solid hydrophobic core and a hydrophilic shell. Various groups described the successful synthesis of thiophene-based conjugated diblock copolymers and their self-assembly to produce well-defined structures at the nanometer scale [35]. N. Elgidawy et al. reported a strategy of self-assembly amphiphilic all-conjugated block copolymers, poly(3-hexylthiophene)-b-poly(3-triethylene-glycol-thiophene), P3HT-b-P3TEGT, consisting of hydrophobic block P3HT and hydrophilic block P3TEGT, into core−shell NPs, as shown in Figure 6. These NPs were further surface coated with mannose to lead to an easy-to-use solution-processable NP material for fabrication of a biosensing layer [36].

Another group described the formation of a CP vesicle with a crystalline core and an amorphous shell using a diblock copolymer of poly(3-hexylthiophene-b-3-(2-(2-{2-[2-(2-methoxy–ethoxy)-ethoxy]-ethoxy}-ethyl))thiophene)P(3HT-b-3EGT) with hydrophobic and hydrophilic side chains [37]. The resulting nanostructured NPs were utilized to investigate how the crystallinity and the optical properties in solution and thin films are affected by the block length. During the self-assembly process, the hydrophobic interactions, which are kinetically or thermodynamically controlled, can be regulated, leading to the creation of either small polymeric micelles or large polymeric vesicles (polymersomes), respectively, by using copolymers with a block of P3HT 26 mol %. They showed that the copolymer can self-assembly at dilute concentrations via a slow dialysis method producing highly ordered polymeric vesicles 200−250 nm in size under thermodynamic control, as shown in Figure 7. By utilizing the polystyrene, the competitive hydrophobic interactions could be used to alter the size of the micelles, while the kinetic control using fast precipitation produced micelles 5–20 nm in size.

### 2.4. Effect of Synthesis Conditions on the Optical Properties of CNPs

CPNs are interesting materials with morphology-dependent optoelectronic properties. In addition, their intrinsic properties (e.g., energy band gap, light absorption and emission, fluorescence quantum yield, etc.) can be tuned through chemical modification. In this section, we discuss the recent strategies to enhance their photophysical properties, fluorescence brightness, fluorescence emission and yield, as well as their fabrication from solution to solid state for future advanced applications. The fluorescence brightness of CPNs, which is proportional to ε × Φ, at the maximum absorption wavelength is an important parameter for imaging applications. However, the fluorescence properties of CNPs are generally limited by quenching phenomena caused by the inter-chain aggregation in the confined NP space. In a recent work, Braeken and coworkers proposed a strategy to enhance the fluorescence brightness of CPNs based on the introduction of branching molecules [38]. Thus, the tri- and tetra-functional branching building blocks were used to construct branched CPNs with an enhanced photoluminescence quantum yield (PLQY). Poly(p-phenylene ethynylene)s (PPEs) were selected as the CP materials not only for their simple structure, but also because of their rigid backbone structure that leads to a relatively high PLQY combined with a large molar extinction coefficient and a high photo-chemical stability. The introduction of acceptor moieties in the PPE backbone allows lowering the band gap energy and obtaining an emission in the NIR region. NPs of all samples were prepared via the combined mini-emulsion/solvent evaporation method. Sonogashira polymerization was used to produce crosslinked CPNs, following the previously published protocols. The influence of the branching molecules was studied; when 3 mol % of any of the branching molecules was used, the copolymers became more gel-like and viscous, and this effect was even more pronounced for 5 mol %. At higher concentrations, 7.5 mol % produced insoluble, likely cross-linked, polymer gel that could not be processed and characterized. The optical properties of the branched polymers in solution and in NP dispersion are investigated and then compared to those of their linear PPE counterparts. The PLQY of the CPNs increase from 5 to 11% for the samples containing 1,3,5-tribromobenzene. The one-photon fluorescence brightness doubles when 5 mol % of either branching molecule is used. The cytotoxicity of branched CPNs was assessed; they show low cytotoxicity in A549 human lung carcinoma cells up to a concentration of 100 μg/mL for 24 h. They also exhibit good NP uptake into cells and compatibility with two-photon imaging.

CPNs that generate a white-light emission when immobilized on a substrate were synthesized by Kim et al. via the Suzuki coupling reaction [39]. To achieve various emission colors with the same excitation wavelength, the phenylene units were added to the polymer’s main chain structure. CPNs with emission colors of red, green, and blue were obtained from the resulting polymers, as shown in Figure 8. Electrospun nanofibers or glass slides coated with such CPNs were used to fabricate white-light-emitting materials under single-wavelength excitation. The benefit of this method is that it avoids the unnecessary energy transfer from a short- to a long-wavelength emission, which is typically known to be challenging when producing a white emission from mixed colors. White light emission was successfully achieved using spatially separated CPNs located on the nanofibers and on the glass.

### 2.5. Effect of Synthesis Process on Morphological and Chemical Properties of CPNs

There are many recent researches focusing on the effect of CNP size and shape, as well as the role of surface chemistry of functionalized CNPs in cellular uptake [40,41,42]. Large NPs can obstruct the trafficking of the targeted biomolecules, limit access to congested cellular sites, and induce steric hindrance, which affects activities including targeting specificity and binding affinity of the attached biomolecules, while, small NPs that range in size from a few to several hundred nanometers enter the cells via pino- or macropinocytosis. It is noteworthy that compared with the inorganic quantum dots, CPNs exhibit less toxicity, which is confirmed by the methyl thiazolyl diphenyl tetrazolium assay [43]. Although there is no comparative study of the toxicity of CNPs and their inorganic counterparts, the low cytotoxicity of CNPs is due to the fact that the CP structures do not contain any toxic elements and they are highly biocompatible. According to Choi et al., NPs with a diameter of less than 6 nm can be eliminated through the renal system, whereas those with a diameter larger than that are primarily absorbed by the reticulo-endothelial system and end up in the liver and spleen [44]. Mendez et al. showed that the subcellular localization and toxicity of CPNs are related to their surface chemistry, the chemical structure and the side chain of the π-conjugated backbone structures. CPNs containing primary amines showed a strong Golgi localization without any toxicity. However, introducing short ethylene oxide side chains and tertiary amine side chains resulted in a decrease of Golgi localization and a highetoxicity, respectively [45]. The study of the biocompatibility of CPNs with relevant blood components was demonstrated by Khanbeigi et al. who investigated three different CPN formulations by varying the type of fluorescent CPs used, the concentration of material and the surface modifying layer. All CPN systems were compared with carboxyl-modified polystyrene beads whose negative surface charge has been proven to activate human platelet cells and influence coagulation pathways. CPNs polyethylene glycol (PEG)-coated surface does not stimulate platelet activation or aggregation, but may cause a small amount of hemolysis in the presence of free surfactant (PEG segments) and can inhibit physiological mediators of platelet aggregation, such as adenosine diphosphate [45]. The same group demonstrated the benefit of the encapsulation of PCPDTBT NPs with biocompatible poly(D,L-lactide-co-glycolide)-poly(ethylene glycol)-PLGA-PEG. When utilizing PEG2kDa-PLGA4kDa, with a content of 5% of PCPDTBT, the mean sizes of NPs were the smallest (<100 nm). The cytotoxicity (IC50 value) of the PEG2kDa-PLGA4kDa system is three times lower than that of the other two systems. No platelet aggregation or inhibition of ADP-induced platelet aggregation was seen, and hemolytic activity was less than 2.5% across all systems. The fluorescence emission shifted toward red wavelengths, and the PLQY dropped compared to the level of the THF-solution. In comparison with the high-molecular-weight polymers, the low-molecular-weight PEG2kDa-PLGA4kDa NPs had the advantages of smaller NP size, less cytotoxicity, and improved imaging performance [46]. In the field of bioimaging, obtaining accurate information about biomolecules is a challenging task [47]. As a result, both the target biomolecules and other extraneous factors such as probe concentrations, excitation power, and cellular context have an impact on the emission wavelength, emission intensity, shape, peak position, or lifetime of fluorescent probes. One approach to address these relies on the design of a single-type of probe with dual emission signals, one to reflect interactions with the target analyte and the other to serve as the internal intensity reference. An example of CPNs with a dual emission property was reported by Yao and Fukui [48].

Fluorescent π-conjugated polyelectrolyte NPs were prepared via the nano-precipitation method. The mean NP sizes were tuned by the net charge ratio between the polyelectrolyte and cation. Using the fluorescence anisotropy approach to investigate the photophysical characteristics of CPNs, including exciton migration, it was observed that a relative increase in efficiency of green-site emission brings a dual emission behavior. As the size of the NPs decreased, the intensity of the green-site emission increased. The non-radiative excitation energy transfer to an emissive trap site, low-energy defect on the conjugated backbone, or the exclusive excitation of the conjugated segments with a long effective conjugation length can all be the causes of the green-site emission. The latter mechanism is important in the small-particle-size region.

A common photophysical processes occurring in CPNs is the excitation energy transfer from the NP to the biological systems or vice versa called excitation energy transfer (EET) or Förster resonant energy transfer which consists in the simultaneous de-excitation of one of the subsystems and the excitation of the other subsystem. EET is one of the key factors that determine the biomedical and bio-imagining applications of CPNs. Due to the excellent funneling of the excitation energy to the trap sites, EET often results in a quenching of the fluorescence in CP materials at length scales of a few nanometers to tens of nanometers. Understanding the relationship between the sizes of the NPs, fluorescence brightness of CPNs, FET has been led to fabricate ultra-small (3.0–4.5 nm) CPNs with excellent photostability through a proper control of the precipitation conditions and the interchain interactions within the CNP [49].

## 3. Functionalization of CNPs for Biomedical Applications

For biosensing and biomedical applications, the surface of the CPNs needs to be functionalized for subsequent bioconjugation to guide the NPs toward a selective target, specific cells or organelles. Such functionalization can be achieved following three approaches: (i) direct functionalization of CPs, (ii) an affinity-driven binding of secondary capping layer strategy and (iii) the last approach, which is an extension of the affinity-driven approach.

### 3.1. Direct Functionalization

This approach relies on the covalent functionalization of the monomer followed by its polymerization. The post-polymerization functionalization approach is also relevant to this strategy. Direct functionalization is the most convenient method to graft functional groups on CP, enabling a direct immobilization of the probe (or targeted biomolecules). The most used organic functions are carboxylic acid, amine or N-hydroxysuccinimide (NHS) groups that allow an easy covalent grafting of targeted biomolecules through the formation of an amide bond. However, the major drawbacks of the direct functionalization strategy lie in its expensive and labor-intensive synthesis processes. For instance, using this strategy, Wang and co-workers prepared many CPNs for fluorescence imaging and clinical diagnosis, with cell imaging and transfection functions, by linking a lipid-modified cationic poly(fluorenylene phenylene) (Lipid-PFPL) and ammonium group [50]. The PFPL NPs showed excellent photostability and little cytotoxicity. These modified NPs can easily enter the cytoplasm via endocytosis (within 4 h) and have the potential to be used for cell imaging. In 2011, the fabrication of cisplatin-bonded CPNs (PT-Pt) was reported. Due to its amphiphilic nature, the copolymer forms NPs in water. Even at a high concentration (100 g mL), the polymer exhibits good photostability and has no harmful effects on human lung epithelial (A549) cells. As a result, this copolymer satisfies the essential criteria for its use as a fluorescent probe for cellular imaging. As a model, cisplatin, the cis-diamminedichloroplatinum(II) molecule widely used to treat cancer, is coupled to polythiophene to create the conjugate “PT-Pt” by coordinating interactions with the amine groups on the side chain of the polythiophene. By using fluorescence microscopy, the PT-Pt can be utilized to track the cellular distribution of cisplatin. The fluorescent imaging platform is provided by amphiphilic copolymers [51].

### 3.2. Functionalization through Affinity Interaction

Regarding the second approach, it involves secondary capping layers that are bound by affinity and depends on the interactions between conjugated chains and a secondary capping agent. In this strategy, the electrostatic and hydrophobic interactions are the driving force to encapsulate the naked CPs chains. The great tendency of naked CPNs to aggregate in water due to the high hydrophobicity of the conjugated backbone causes an unintentional self-quenching of fluorescence. To overcome this issue, surface modification with PEG is a desirable option which improves hydrophilicity and biocompatibility. Most importantly, the PEG chain’s length can be controlled to accommodate various biological applications, which is crucial. Additionally, PEG is a useful functional polymer for biological applications due to its ease of modification with certain functional groups (amino or carboxyl groups, lipid, and biomolecules). For fluorescent imaging, multicolor regulation, gene delivery, and medication delivery and release, several PEG-functionalized CPNs have been reported. For instance, recently, we described the self-assembly of P3HT-b-P3TEGT into core–shell nanoparticles which were then mannose-decorated. The amphiphilic behavior of P3HT-b-P3TEGT helps it to form nanoparticles in water or methanol, in which hydrophobic moieties come together to form the inner core and the PEG segments on the outer surface of the NPs. Using non-covalent and hydrophilic–hydrophilic interactions, mannose was coated during the formation of NPs, as shown in Figure 9 [36].

In order to enhance the NPs hydrophilicity and the incorporation of anchoring sites for subsequent functionalization for biosensing, imaging, and treatment applications, Wu et al. reported CPNs modified by an amphiphilic copolymer, poly(styrene-alt-maleic anhydride), that was adsorbed on the surface of NPs [52]. In the same idea, Green and co-workers reported the elaboration of multicolor CPNs by encapsulation of commercial CPs within the PEG-phospholipid layer [53]. Their protocol is simple and reproducible. The resulting NPs exhibited high colloidal stability and outstanding optical properties. Likewise, Christensen et al. combined CP based on PFBT with lipid to create CPNs with enhanced brightness and PLQY up to 46%. In addition to phospholipid or PEG encapsulation approaches, functional CPNs can also be produced by electrostatic interactions between a cationic CP and an anionic functional group [54]. By combining cationic PFO and poly(L-glutamic acid) modified anticancer drug doxorubicin (PG-Dox), bifunctional CPNs with the ability of both drug transport and imaging activities have been reported [55]. Moreover, cationic PBF polymers and negatively charged SDPA have been assembled to form multifunctional CPNs with light-activated anticancer and antibacterial activity, as well as imaging capability. The distinctive characteristics of the second approach compared to the first are its simplicity in the functionalization and the use of commercial availability CPs [34].

### 3.3. Multi-Functionalization Approaches

The third design strategy can be considered as an extension of the second one. In such approach, the surface of CPNs is first functionalized with a carboxyl or an amino group, as in the second approach. The functional biomolecule (peptide, sugar, protein, antibody) is then covalently bonded to the surface of the nanoparticle via carboxyl condensation or bio-orthogonal click reactions. Based on bioconjugation method, Chiu and co-workers developed functional CPNs for labeling cellular targets through the antigen–antibody interaction or the biotin–streptavidin system [56]. In their work, the authors functionalized CPNs by entrapping polymer chains into a single nanoparticle driven by hydrophobic interactions during nanoparticle formation. An amphiphilic polymer bearing functional groups is co-precipitated with a conjugated polymer to modify the nanoparticle surface for subsequent covalent functionalization with biomolecules, such as streptavidin and immunoglobulin. In this approach, the functionalization and bioconjugation models are almost applicable to all kinds of hydrophobic fluorescence CPs. Later, the same approach has been used by others to prepare a new bioimaging probe based on poly(9,9-dihexylfluorene-alt-2,1,3-benzoxadiazole) (PFBD) NP surface functionalized with PEG-mono-carboxylate. The carboxylate-PEG functionalized NPs have several desirable structural and photophysical properties: a high colloidal stability in a broad range of pH values, an outstanding PLQ of 46%, and the presence of a carboxylic group as a surface chemical functionality. PEG-NPs were then bioconjugated with a cyclic RGDfK targeting peptide for labeling of membrane aVb3 integrin receptors on live HT-adenocarcinoma cells [57].

## 4. Biomedical Applications

### 4.1. Antimicrobial Properties of CPNs

CPs have attracted a great deal of interest as emerging materials in photodynamic therapy (PDT) due to their high light-harvesting capability and efficient energy transfer. They possess a large π-delocalized electronic system where the exciton can migrate along the polymer backbone chains or hop between chains. The ability of the exciton to diffuse throughout CP chains enables efficient energy transfer to dye acceptors such as photosensitizers (PSs) [58]. Moreover, CPs upon photo-excitation can directly sensitize oxygen molecules to produce reactive oxygen species (ROS) which can strongly damage biomacromolecules for antimicrobial and anticancer applications [59]. Whitten and co-workers reported a cationic conjugated polyelectrolyte derivate with the irradiation of visible light which can kill Gram-negative as well as Gram-positive bacteria [60]. This pioneer work opened up a new field of research of CP in the photodynamic therapy. Since then, various conjugated polyelectrolytes and hydrophobic CPNs have demonstrated their utility for PDT. Regarding the biological and biomedical applications of CPNs, the Wang group has reviewed in detail the development of CPNs for fluorescence imaging, anti-microorganism and antitumor, and gene delivery and drug delivery/release. Since then, various families of CPNs have been developed very fast and prosperously from polymer design to applications. Besides PDT, CPNs have recently emerged as a novel anti-bacterial material with a remarkable activity and are considered promising candidates to overcome the bacterial resistance with enhanced efficacy by two different modes of activity, under light illumination and in the dark.

#### 4.1.1. Light-Activated Mode Activity

PDT is a promising method not only for cancer treatment and but also for antibacterial treatment that involves PSs and light irradiation to produce reactive toxic ROS from intracellular oxygen (O_2_) to kill bacteria. Compared to antibiotics, PDT has a number of benefits, including an effective therapeutic impact, few side effects, and negligible drug resistance [61]. In a PDT process, the ROS is produced by irradiating a PS to sensitize oxygen. As a result, the ability of PS to produce ROS has an impact on the PDT’s effectiveness. When the right molecular design is used, CPs could have superior light-harvesting capacity and efficient intra- and intermolecular energy transfer, which can result in a notable increase in ROS generation efficiency compared to standard PSs. The ability of conjugated polymers to sensitize oxygen to generate singlet oxygen and other ROS is due to their aptitude to strongly absorb visible light and to their outstanding fluorescence properties with a high yield of triplet state formation. Both light-activated modes of microbial killing or cancer treatment involve the generation of ROS. This is because of their exceptional fluorescence features, which include a high yield of triplet state production and, as a result, a high generation of singlet oxygen and ROS, as well as their capacity to significantly absorb visible light. The production of ROS is a necessary component of both light-activated cancer treatment and microbial eradication [34]. Luckily, CPs have good light-harvesting and energy transfer properties in comparison to other current PSs with poor solubility and small absorption cross-section [3]. In the last decade, different approaches combining common PSs with CPs to fabricate various CPNs have been developed to advance the improved PDT. By virtue of electrostatic and ligand–receptor interactions, positively charged specific elements can be associated with cells because cell membrane surfaces are negatively charged and have many specific markers. Exploiting this property, Wang and co-workers designed cationic multifunctional CPNs with light-activated anti-bacterial activity with imaging capability [34]. These positively charged CPNs opened up a new avenue in the design of PDT materials. Whitten and co-workers developed different CP systems to disclose and verify the biocidal mechanism of CPs by studying the effect of (i) positively charged CPs which effectively capture negatively charged microbial pathogens by electrostatic and hydrophobic interactions; (ii) singlet oxygen and/or other ROS generated upon irradiation of the CPs; (iii) ROS damaging effect of the membrane of pathogens and killing pathogens [62]. Due to easy synthesis and efficient ROS generation, CPNs are the most popular PDT material. Zhao et al. designed a biocompatible and biodegradable e-poly-Llysine (EPL)/poly (e-caprolactone) (PCL) copolymer [63]. This amphiphilic copolymer showed good ability to form monodispersed NPs, with a broad-spectrum antibacterial activity against Escherichia coli, Staphylococcus aureus and Bacillus subtilis. It is worth noting that the new NPs had a stronger antibacterial activity than the cationic peptide EPL. Several possible antibacterial pathways have been proposed in order to investigate the underlying mechanism of the biodegradable cationic NPs. It has been found that NPs can damage bacterial walls and membranes and cause an increase in ROS levels and alkaline phosphatase. More importantly, the self-assembled NPs changed bacterial osmotic pressure, resulting in cell invagination and formation of holes which cause a leakage of the cytoplasm. All these points show that the EPL–PCL NPs can be further developed to be a promising antimicrobial material for infectious disease treatment or as surfactants and emulsifiers to enhance drug encapsulation efficiency and antimicrobial activity, as shown in Figure 10.

Zhang et al. demonstrated dual-mode antibacterial CPNs by combining photothermal therapy and PDT. the authors explored them for efficient killing of ampicillin-resistant *Escherichia coli* (Ampr *E. coli*), as shown in Figure 11 [64]. 

The CPNs with a size of 50.4 ± 0.6 nm for dual-mode phototherapy were fabricated by co-precipitation of poly(diketopyrrolopyrrole-thienothiophene) (PDPPTT) and the PS poly [2-methoxy-5-((2-ethylhexyl)oxy)-p-phenylenevinylene] (MEHPPV) in the presence of poly(styrene-co-maleic anhydride). The latter polymer makes NPs disperse well in water via hydrophobic interactions. Thus, the dispersion of these NPs has a photothermal effect and a bacterial killing ability, simultaneously, since NPs are sensitive to the oxygen in the surrounding environment and generate ROS upon the illumination of light. With a combination of near-infrared light (550 mW cm^−2^, 5 min), white light (65 mW cm^−2^, 5 min) and a concentration of 9.6 × 10^−4^ μm of NPs, one could reach a 93% inhibition rate against Ampr *E. coli*, which is higher than the efficiency treated by PTT or PDT alone. These NPs with a dual mode offer the possibility of treating pathogenic infections brought on by resistant microorganisms in a clinical setting.

Wang et al. proposed a facile and rapid photothermal antimicrobial platform based on CPNs functionalized with the cell-penetrating peptide CPNs-Tat that have a near-infrared (NIR)-active and photothermal-responsive ability, as shown in Figure 12 [65]. Due to the positively charged Tat peptide, CPNs-Tat can enhance the interaction with bacteria cells by forming a CPNs-Tat/bacteria aggregation. Under NIR irradiation, CPNs-Tat could efficiently convert light into heat and create local hyperthermia to quickly kill bacteria. This photothermal approach provides an expedient and efficient tool for eradicating bacterial infections.

Chong et al. synthesized water-soluble cationic CPNs based on polymer containing PBF repeat units in the backbone with negatively charged SDPA in an aqueous solution through electrostatic interactions [34]. Upon photoexcitation of the NPs with white light (400–800 nm) with 90 and 45 mW cm^−2^, they rapidly kill bacteria and cancer cells through the generated ROS. Additionally, these PBF NPs simultaneously offer optical imaging and bacteria-killing capability. PBF NPs are thus a promising multifunctional material for the simultaneous provision of optical monitoring capabilities and the treatment of bacterial infections and malignancies, as shown in Figure 13.

#### 4.1.2. Dark Mode Activity

The mechanism of antimicrobial activity of CPNs is still under investigation. It may involve membrane destruction by physical/chemical properties of NPs, as do many antibacterial peptides. Some recent works have demonstrated that naturally occurring antimicrobial peptides and their synthetic counterparts act only when in contact with bacterial cell walls by targeting the lipid bilayer of the bacterial membrane. The ability of these molecules to perturb the bacterial cell is highly dependent on the lipid composition of the membrane [66]. Furthermore, most antimicrobial peptides are cationic. Therefore, both the light-activated and dark biocidal activities of cationic CPNs are linked to their interactions with bacterial cell membranes. The dark killing abilities of these compounds correlate with their perturbation ability in relation to bacterial membranes. Charge density is important for the initial binding step between antimicrobial compounds and bacteria [67,68,69]. Recently, cationic fluorescent CPNs were shown to exhibit a broad-spectrum dark antimicrobial activity. Outstandingly positively charged cationic CPNs can physically combine with bacteria during bacterial incubation, then destroying the bacteria membrane and killing the bacteria in the dark, as shown in Figure 14 [70]. Then, NPs could diffuse through the cell wall and irreversibly disrupt the membrane structure of bacteria, which leads to the release of the cytoplasmic constituents from cells and bacteria death.

In our previous work, we described a facile and cost-effective approach for preparing water-based cationic CPNs (CTAB-P3HT NPs) [71]. The combination of cationic surfactant cetyltrimethylammonium bromide (CTAB) with neutral CP (P3HT) provides positively charged NPs that not only allow electrostatic interaction with bacteria, but also synergize the antimicrobial activity of CP in the dark. The prepared cationic CPNs (CTAB-P3HT NPs) showed a broad spectrum of anti-bacterial and antifungal activity without requirement of light with a good biocompatibility toward normal human cells. This work provides a new biocompatible theragnostic NPs for bacteria detection as well as for the therapeutic treatment.

### 4.2. Fluorescence Sensing

In recent years, CPNs have emerged as exceptional fluorescent probes for in vitro and in vivo bioimaging and biosensing applications because of their exceptional optical properties combined with high fluorescence brightness, good photostability, and lower toxicity [1,3]. CPNs are made of conjugated polymers with π-delocalized electronic structures with distinct emission colors, a narrow bandwidth and are strongly fluorescent. This latter property, together with an easy preparation, high fluorescence brightness, low toxicity, stable photoluminescence and small particle size, makes them promising materials as fluorescent probes and fluorescent biosensors. As discussed in the previous section, CPNs are easy to prepare and their optical properties as well as their size can be easily tuned.

The most important examples of fluorescent CPs developed are compiled in Table 1 and Table 2. These tables compile key parameters on physical properties of these nanoparticles. The representative family of CPs includes polyfluorene (e.g., PDHF and PFO), poly(phenylene ethynylene) (e.g., PPE), poly(phenylenevinylene) (e.g., MEH-PPVand CN-PPV), fluorene-based copolymers (e.g., PFPV, PFBT, and PFDBT), poly(thiophene), and related derivatives, which manufacture NPs with emission colors spanning the whole range of the visible spectrum.

On the other hand, fluorescent-based biosensor is a widely used approach for quick and easy medical diagnosis due to its inherent sensitivity and high selectivity [72,73]. This has stimulated enormous research and development of highly fluorescent π-CP as an active material in light-emitting devices and as a sensing layer in biosensor devices [74]. Thus, various fluorescent biosensing materials have been developed based on fluorescent π-CP for sensing metal ions, small molecules to microorganisms and cells [75,76,77]. The detection mechanism could be achieved by two approaches, by quenching the fluorescence of the polymers after the reaction with the target or by the emission of fluorescence.

#### 4.2.1. Detection through Fluorescence Quenching

Wu et al. explored the size-controlled preparation of several new CPNs that show remarkable high-fluorescence quantum yield, radiative rate, photostability, high effective chromophore density, and minimal levels of aggregation-induced fluorescence quenching, resulting in fluorescence quantum yields exceeding 70%, even in the solid-state, thin films [78]. Other advantageous features of these CPNs include the lack of small dye molecules and heavy metal ions that might dissolve into a solution. Based on the CPN approach, Yuan et al. proposed a straightforward and accurate fluorescent technique for hydroquinone detection [79]. They characterized their CNPs by TEM, DLS, zeta potential, and FT-IR. The possible quenching mechanism may be caused by the electron transfer from hydroquinone to CPNs that turn off the fluorescence, as shown in Figure 15, since CPNs catalyzed the oxidation of hydroquinone to benzoquinone which could effectively quench the fluorescence of CPNs. This result was confirmed through the incubation of CPNs and the hydroquinone mixture with glutathione which induced a fluorescence recovering due to the strong antioxidant property of glutathione, so the benzoquinone was reduced into hydroquinone. The detection limit of hydroquinone down to 0.005 μM with excellent selectivity was achieved. This result of real sample analysis showed that this approach could be used in a real application. The proposed biosensor is expected to be beneficial for tracking the level of hydroquinone in the environment.

Recently, in our work, we reported a fluorescence-sensing scheme based on CTAB-P3HT NPs for the detection of the E. coli in the concentration range of (50–0.5 × 10^6^ CFU/mL) with a limit of detection (LOD) less than 5 CFU/mL using a simple mix and detect strategy, as shown in Figure 16 [71]. As seen in Figure 16a, the interactions between positively charged CTAB-P3HT NPs and hole microbial particles result a decrease in fluorescence intensity. The aggregation of CTAB-P3HT NPs on E. coli, which results from the electrostatic interactions between the cationic groups of CTAB-P3HT NPs and a negative charge on the bacterial cell wall, can be used to explain the fluorescence quenching.

Li et al. reported a ratiometric fluorescent probe based on coordination CPNs for direct and fast detection of Zn^2+^ in aqueous solutions with excellent stability and selectivity [80]. The probe was prepared from aggregation-induced emission (AIE) fluorophore (4,4′-(hydrazine-1,2-diylidene bis (methanylylidene)) bis (3-hydroxybenzoic acid)) (HDBB) molecules with metal ions. The CPNs are made up of HDBB molecules and Tb3+ (referred to as Tb-HDBB-CPNs), which exhibited a matrix coordination-induced emission peak at a wavelength of 590 nm. In contrast, Zn-HDBB-CPNs exhibited a distinctive fluorescent property with a blue emission peak wavelength of 470 nm (Figure 17). Based on the cation exchange process of Tb HDBB-CPNs with Zn^2+^, a highly selective ratiometric fluorescent probe in the range of 100 nM to 60 mM and a detection limit of 50 nM was developed for the determination of Zn^2+^ in an aqueous solution. This research demonstrates the benefits of using AIE materials as a fluorescent probe and exhibits the advantages of a direct detection procedure. Thus, it paves a way and opens up the avenue for further research in the AIE functionalized materials with ratiometric fluorescence response for chemical sensing.

#### 4.2.2. Sensing through Fluorescence Emission

Zhang et al. demonstrated a novel and efficient melamine biosensor based on fluorescence FRET between CPNs and AuNPs, as shown in Figure 18 [81]. The system is highly selective and sensitive to the detection of melamine, with a LOD of 1.7 nmol/L. The energy transfer efficiency can reach 82.1%. The interaction mechanism occurred when the CPNs were able to close the surface of the AuNPs. Then, the fluorescence of CPNs was effectively quenched. However, the aggregation of AuNPs by the N-Au interaction after melamine addition reduced FRET of CPNs-AuNPs and promoted the recovery of CPN fluorescence. After the addition of melanin, the absorption spectrum Au-NPs overlapped significantly with the emission spectrum of the CPNs, which reduced the energy transfer efficiency. Furthermore, the proposed technique was carried out for melamine detection in real powdered infant formula with satisfactory results.

Wang et al. described a fluorescent probe for sulfide based on carboxylic functionalized CPNs [82]. The NPs were manufactured by the co-precipitation method from poly[(9,9-dioctylfluorenyl-2,7-diyl)-co-(1,4-benzo-2,1′-3-thiadiazole)] bearing carboxylic functional group and referred to as COOH-PFBT. The green fluorescence of these carboxylic CPNs is quenched due to the aggregate upon the addition of Cu(II) ions. This quenching process is fully reversible; thus, the fluorescence emission is restored when the sulfide is added to the aggregates, inducing the formation of CuS. This quenching recovery (Boff-on^) mechanism forms the basis for a new sulfide detection scheme. The fluorescence intensities vary linearly with the concentration of the sulfide in the range of 1.25 to 75.0 μM, with a LOD of 0.45 μM. This biosensor has a good selectivity for sulfur anions over other anions. When used for the determination of sulfide in spiked real water samples, this method demonstrates recoveries ranging between 98.6% and 105.7%.

#### 4.2.3. Fluorescent CPNs for Bioimaging

Bioimaging is crucial in the field of medical diagnosis, biological and biomedical research to gain insights into biological processes and malfunctions in many ways, including the observation of morphology and structure, study of functions, and diagnosis and therapeutics of diseases. Some of the notable progress made in this field includes the development of various probe materials with outstanding properties for cell imaging, subcellular imaging, molecule imaging, and so on. Besides probe material development, progress in the fluorescence imaging techniques has become a prevalent choice providing effective research tools to investigate many fundamental processes in the life sciences. It is widely used in a broad variety of preclinical investigations for the visualization of biological processes and provides cellular and subcellular resolution and image enhancement in 3D and real time in a non-destructive way. As a result, fluorescence-based diagnosis of diseases and fluorescence-image-guided surgery found many real applications in human disease diagnosis [3,83]. Nevertheless, the future search for improved cost-effective, time-dependent and safe bioimaging techniques with excellent resolution is still ongoing. Fluorescent NPs have proven their excellent intrinsic properties including their resonator geometry, which allow amplified emission from a single fluorescent NP. This latter property is of interest for tracking of individual cells in larger united cell structures. Whispering-Gallery-Mode-based biosensors (WGMS) can be categorized as label-free technologies. Their basic principle relies upon the total internal reflection of light trapped inside a circular resonator. Certain circular nanoparticles satisfy the conditions of biosensing with WGMs mode lasers. Among these criteria, the NP diameter needs to be in the range of several micrometers to support the lasing modes. However, with these sizes, the NPs are less important for in vivo applications; however, there might be solutions to make a single particle laser resonator smaller by confining the fluorescence and gain to subwavelength surface plasmon polaritons.

Traditional fluorescent probe materials for imaging such as fluorescent organic dyes have excellent fluorescence properties, including high PLQYs, and a large variety of these dyes with tunable optical characteristics are readily available. However, organic dyes suffer a low photo stability and light bleaching issue. Inorganic quantum dots overcome these limitations; however, they contain heavy metals like lead, cadmium, or indium, which significantly increase their cytotoxicity. Despite the recent progress in this filed, cytotoxicity and photo-bleaching issues may significantly restrict their use in real applications [84,85]. In the last decade, CPNs have surpassed conventional organic dyes and become a new class of probe materials for bioimaging due to their excellent photostability and high fluorescence brightness. Alongside with their applications in imaging, CPNs are appealing for cell culture research because of their beneficial intrinsic properties such as low cytotoxicity, inertness to intracellular processes, and their high resistance to bleaching and easy chemical functionalization [83,86]. For potential in vivo applications, CPNs should meet certain requirements including an ideal size range, as they are small enough to allow access to smaller capillary blood vessels and large enough to not be cleared immediately, which impacts therapeutic strategies, surface charges, cell viability and the associated pharmacokinetics. Both non-specific and specific imaging can be targeted by CNPs. Non-specific imaging relies on the direct use of CPNs without functionalization with a specific targeting element. Many prior studies have mainly focused on non-specific cellular imaging of CPNs. This mechanism is driven by non-specific cellular uptake, and CPNs are mainly located in the cytoplasm. Therefore, the biocompatibility and low cytotoxicity are the main criteria for assessing the imaging performance of CPNs. Additionally, CPNs are also a new class of probe materials for theranostic agents [87]. As an example of application, CATB-P3HT-NPs that were designed by our group [71] were used to image bacteria by fluorescence confocal microscopy. The bacteria emit bright red fluorescence, indicating that NPs with positive charge are bonded to bacteria via electrostatic interactions, as shown in Figure 19.

Tang et al. developed fluorescent CPNs based on the model CPs, poly (fluorene divinylenebenzothiadiazole) PFVBT, and the model quarternary ammonium salts, cationic surfactant CTAB. These CA-CPNs were successfully used in the imaging of both bacterial cells and mammalian cells with selective toxicity against bacteria, as shown in Figure 20 [70].

Recently, other innovative CPNs have been designed and synthesized for various modal imaging techniques, as shown in Table 1 and Table 2.

**Table 1 microorganisms-11-02006-t001:** Chemical names and corresponding abbreviations of conjugated polymers and nanoparticle stabilizing agents.

Abbreviation	Chemical Name	Ref.
Conjugated polymers		
DPA-CNPPV	(Poly[{2-methoxy-5-(2-ethylhexyloxy)-1,4-(1-cyanovinylenephenylene)}-co-{2,5-bis(N,N′-diphenylamino)-1,4-phenylene}]	[88]
(APNs)	Polyfluorophore Nano sensors	[89]
PBTQ4F	Poly[4-(4,8-bis(5-(2-ethylhexyl)thiophen-2-yl)benzo[1,2-b:4,5-b′]dithiophen-2-yl)-2-(2-ethylhexyl)-6,7-bis(3-((2-ethylhexyl)oxy)-4,5-difluorophenyl)-2H-[1,2,3]triazolo[4,5-g]quinoxaline]	[90]
(PTZTPA-BBT)	Triphenylamine (TPA) functionalized phenothiazine (PTZ) as the donor, strong acceptor benzothiazole (BBT).	[91]
DPP-BTzTD	Poly([2,5-bis(2-decyltetradecyl)-2,5-dihydropyrrolo[3,4-c]pyrrole-1,4-dione-3,6-dithienyl]-co-[6-(2-ethylhexyl)-[1,2,5]thiadiazolo [3,4-f]benzotriazole-4,8-diyl])	[92]
SPNRs (PFPV, PFBT, and PFODBT)	Semiconducting polymer nano reporters	[93]
PFPV	Poly[(9,9′-dioctyl-2,7-divinylene-fluorenylene)-alt-{2-methoxy-5-(2-ethylhexyloxy)-1,4-phenylene}]	[93]
PFBT	Poly[(9,9′-dioctylfluorenyl-2,7-diyl)-alt-(benzo[2,1,3]thiadiazol-4,7-diyl)]	[93]
PFODBT	Poly[2,7-(9,9′-dioctylfluorene)-alt-4,7-bis(thiophen-2-yl)benzo-2,1,3-thiadiazole	[93]
PBMC	Poly[3-{2-[2,5-Bis-(2-ethyl-hexyloxy)-4-propenyl-phenyl]-vinyl}-9-butyl-6-methyl-9H-carbazole]	[94]
PTD	Poly[6-(2-ethylhexyl)-4-methyl-8-(5-methylthiophen-2-yl)-6,7-dihydro-5H-[1,2,3]triazolo[4′,5′:4,5]benzo[1,2-c][1,2,5]thiadiazole]-alt-[3-methyl-6-(5-methylthiophen-2-yl)-2,5-bis(2-octyldodecyl)-2,5-dihydropyrrolo[3,4-c]pyrrole-1,4-dione]poly-thiadiazolobenzotriazole-alt-thiophene-diketopyrrolopyrrole	[95]
PCPDTBT	Poly[2,6-(4,4-bis-(2-ethylhexyl)-4H-cyclopenta[2,1-b:3,4-b′]-dithiophene)-alt-4,7-(2,1,3-benzothiadiazole]]	[96]
PDFT	Furan-containing diketopyrrolopyrrole-based semiconducting polymers	[97]
P3HT	Poly(3-hexylthiophene)	[71]
NP stabilising agents (surfactants/polymers/lipids)		
PSMA	Poly(styrene-co-maleic anhydride)	[88]
(PS-PEG-COOH)	Polystyrene graft ethylene oxide functionalized with carboxylic end group	[90]
PEG-b-PPG-b-PEG	Triblock surfactant of PEG and poly-(propylene glycol) (PPG)	[93]
DSPE–PEG	1,2-Distearoyl-sn-glycero-3-phosphoethanolamine-N-[methoxy(polyethyleneglyol)]	[95]
PEG-PLGA	Poly(ethylene glycol)methyl ether-block-poly(lactide-co-glycolide) copolymer	[96]
PEG	Polyethylene glycol	[97]
CATB	Cetyltrimethylammonium bromide	[71]

**Table 2 microorganisms-11-02006-t002:** Polymer and stabilizing agents used in CPNs, their size, PLQY (%) and applications, as reported in the literature.

Conjugated Polymer	CPN Stabilizing Agent	Size (nm)	PLQY (%)	Application	Ref.
DPA-CNPPV	PSMA	17.5–22.1	10.8	NADH sensing	[88]
APNs	Protease-reactive peptide brush (via self-immolative linkers)	100–200	N.A.	Cancer and allograft	[89]
m-PBTQ4F	(PS-PEG-COOH)	22	3.2	Fluorescent imaging	[90]
Pdots-C6 (PTZTPA-BBT)	(PS-PEG-COOH)	40.8	N.A.	Fluorescent imaging	[91]
DPP-BTzTD	Self-assembling conjugated polymer	4	N.A.	Photoacoustic imaging	[92]
SPNRs	PEG-b-PPG-b-PEG	30–40	2.7 ± 0.014	Chemiluminescent imaging	[93]
PBMC	PSMA	∼44	N.A.	Fe^3+^ sensing in vitro in HeLa cells	[94]
PTD	DSPE–PEG	42	N.A.	In vivo photoacoustic 3D vasculature imaging	[95]
PCPDTBT	PEG-PLGA	32 ± 0.7	2.8	Fluorescent imaging	[96]
PDFT	PEG	40	N.A.	Fluorescent imaging	[97]
PBTQ4F and CN-PPV	PSMA	∼15	1.9 and 6	Fluorescent imaging	[98]
Fluorinated PBTQ	(PS-PEG-COOH)	≈22	3.2	Fluorescent imaging	[90]
Pttc-SeBTa-NIR1125/1270/1380	DSPE–PEG	35–73	0.05–0.18	Fluorescent imaging	[99]
P3HT	CATB	55.32 ± 7.3	N.A.	Fluorescent imaging, bacteria sensing, and killing	[71]

### 4.3. Electrochemical Sensing

Recently, intensive studies were carried out based on CPNs and their composites for electrochemical sensor development, for the highly sensitive detection, their porosity, large surface area, and the presence of binding sites originating from their functional groups for enabling biomarker capture as well as ultrasensitive and easy detection. In addition, due to their ability to convert a biological interaction into a measurable electrochemical signal, they became a material of choice for sensing applications. Consequently, CNPs can act as excellent electrode materials for immobilizing various molecular recognition elements through the most common covalent attachment method and the often-used entrapment or adsorption methods, resulting in ultrasensitive recognition with very low LOD. More importantly, functionalizing the surface of CNPs has been found to play a major role in limiting the biofouling effect, resulting in ultrasensitive detection in real samples. This anti-fouling effect is a supreme character for biomarker detection in real practical applications [100].

As a result, the fabrication of CPN-based biosensors with ultrasensitive, selective and antifouling effect is expected to be a promising diagnostic tool that can replace the conventional methods for analysis. However, a lot of future work is highly needed for the development of new CPN-based biosensors since they still have many unexplored possibilities. Another promising method for the future is the development of CPN-coated nanowires or nanotubes which recently highly improved the sensing properties of CPNs [101]. In addition, designing hybrid conducting polymer nanocomposites based on polymers and conducting inorganic nanomaterials such as carbon nanomaterials and metallic nanoparticles to produce CPNs nano-systems will result in a high-potential and synergistic effect with promising different materials, yet to be explored for biosensing.

While using conducting polymers based electrochemical sensors, the capture of a target analyte generates a measurable signal which is directly converted into an electrical signal, as shown in Figure 21. The efficiency of the biosensor depends on the strength of the bioreceptor immobilization to the conducting polymer and of the conducting polymer to the biosensor surface. This accordingly affects the selectivity of the biosensor that depends on the specific interactions between the analyte and the bioreceptor. The sensitivity of the biosensor is determined by the electrochemical signal intensity resulted from the reaction between the analyte, the bioreceptor and the conducting polymer. This electrochemical signal may result from a change in the voltage, current, conductivity/resistance, impedance, or number of electrons exchanged through an oxidation or reduction reaction, leading to the development of potentiometric, amperometric, conductimetric, impedimetric and voltametric biosensors, respectively [102].

Shekher et al. [103] recently demonstrated the fabrication of a biosensor for a sensitive detection of serotonin (5-HT) based on electrodeposition of 2-amino-5-mercapto-1,3,4-thiadiazole (AMT) and gold nanoparticles (nAu) on the functionalized carbon nanotube (f-CNT) on a glassy carbon electrode. An LOD of 7.8 nM was achieved by using cyclic voltammetry, square wave voltammetry, and i-t amperometric analysis. For practical utility, it was investigated using human serum with good recovery limits of 96.3 to 103.6%. More recentlym, Tseng et al. [104] investigated the electrochemical detection of Leukocyte esterase (LE) and nitrite (NIT) which are useful biomarkers for the diagnosis of urinary tract infections (UITs). A carboxylic acid-functionalized poly(3,4-ethylenedioxythiophene) (PEDOT-COOH)-modified carbon electrode was used to immobilize the LE antibody and fabricate the biosensor. Different electrochemical techniques such as differential pulse voltammetry and electrochemical impedance spectroscopy were used to characterize the biosensor. Good performances with LODs of 6.24 µM and 0.2 μg L^−1^ in a linear range of 9.1–131 µM and 0.2–590 μg L^−1^ were demonstrated for NIT and LE, respectively. In real urine specimens, a diagnostic sensitivity and specificity were 100% and 87.5%, respectively. Another novel hydrogel-based electrochemical biosensor was recently developed by Du et al. [105]. The biosensor was fabricated from the antifouling zwitterionic peptide hydrogel (CFEFKFC), modified with the conducting polymer poly(3,4-ethylenedioxythiophene) (PEDOT) and gold nanoparticles (AuNPs) through electrodeposition to detect prostate specific antigen (PSA) in human serum. The hydrogel was immobilized onto the electrode surface via the Au–S bond. It is also suitable for the immobilization of PSA antibodies through the formation of covalent amide bonds. The outstanding antifouling property of the hydrogel could effectively prevent the adsorption of nonspecific proteins, cells and other biomolecules. The results showed a linear range from 0.1 ng mL^−1^ to 100 ng mL^−1^, with a low (LOD) down to 5.6 pg mL^−1^ [105]. Kumar A et al. developed an electrochemical biosensor based on a graphene oxide/polyaniline/polypyrrole/zinc oxide (GO/PPy/PANI/ZnO) nanocomposite, NC, for cholesterol and bilirubin detection. The results showed a sensitivity of 0.92 μA μM^−1^ cm^−2^ and 0.2 μA μM^−1^ cm^−2^ for the detection of cholesterol and bilirubin, respectively. In comparison to the existing reports, the detection sensitivity for cholesterol is almost doubled, and for bilirubin, it is twenty times greater. The sensor showed remarkable detection limits, which were 0.42 and 0.69 μM for cholesterol and bilirubin, respectively [106]. There is a series of examples of CP-based electrochemical biosensors summarized in Table 3 to highlight the recent advances in this field.

## 5. Conclusions and Perspectives

CPNs have become an important emerging material for in vivo bioimaging and sensing, which greatly support the disease diagnosis and treatment. In this review, we survey the current state of the art of methods of preparation, advantages and disadvantages, as well as bio-applications of CPNs. Several methods of CPN preparation are discussed to demonstrate the effect of synthesis conditions on the optical, morphological, and chemical properties of CPNs. CPNs are widely used in biomedical applications due to their unique photophysical and photoelectric properties that endowed them excellent performance with high quantum yield, photostability and favorable compatibility. In addition, controllable particle size can be acquired, and these particles can easily be functionalized to be adapted to specific application requirements. Considering all the unique properties of CPNs mentioned in this review, we suggest a superior utility for practical applications compared to that of the conventional materials. As biosensors, CPNs could efficiently detect glucose, cholesterol, serotonin (5-HT), phenylalanine, prostate specific antigen, urinary tract infections and tumor markers both in buffers and human serum with excellent, rapid and accurate results in a cost-effective and eco-friendly way. CPNs are also superior for biological applications such as fluorescence, luminescence, Raman, and PA imaging due to their unique photophysical properties, biocompatibility and rapid metabolism relevant for biosafety compared to other imaging agents. Such an outstanding performance with excellent results in various biomedical applications has confirmed the significance of CPNs for bioimaging, drug delivery, and sensing applications. Regarding biomedical applications, the superior brightness of CPNs and their photostability as well as their low toxicity permits the tracking of subcellular entities, or labeled molecules over extended periods of time with high resolution. Although there are many research reports available in the open literature; however, the field is still in its infancy, and much of the effort so far has been placed on the development of new conjugated polymer structures as nanoparticle materials and their photophysical characterization. To proceed further in this field, biomedical application is required, and not just proof-of-concept study. To progress in this direction, organic and material chemists need to work side by side with biologists. CPNs have been used in diagnostic techniques with great achievements in the sensing and imaging markets; however, their clinical applications for the diagnosis of diseases and treatment are still in the research stage due to several aspects that should be taken into consideration in the future development of CPNs for biomedical applications. The major challenges facing the use of CNPs in the biomedical field can be addressed as follows:(1)The design of novel CPNs with controllable dosages and unique properties is needed to effectively achieve the goals such as improving ability of photoenergy transfer to lower the side effects.(2)Improved targeted drug delivery is highly needed for cancer therapy through studying the attachment of targeting ligands to the CPN surface.(3)Further investigation of CPN biodegradation time is required to confirm its biosafety in the human body through carrying out a pharmacokinetic study upon injection of CPNs into the animal model.(4)More efforts are still needed to make CPNs scalable through cost-effective and eco-friendly preparation methods such as a cradle-to-grave process study described in this review.(5)Functionalization of CPNs through optimized strategies to improve their performance in biological applications is suggested such as stimulus-responsive targeting, for example, pH-responsive polymers.(6)Synthesis of ultra-small-size CPNs is needed through optimized preparation methods to enhance its biodistribution and achieve a rapid metabolism; their current state currently limits the clinical application of nanoparticles.(7)More efforts should be made to design nanocomposites based on CPNs for multifaceted applications.(8)The rapid advancement in artificial intelligence and computational simulations can be exploited to design functionalized rational CPNs for biomedical applications. Overall, more efforts are highly required to bring CPNs into real clinical use.

## Figures and Tables

**Figure 1 microorganisms-11-02006-f001:**
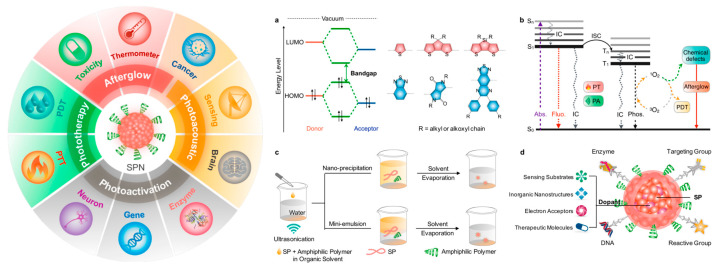
(**Left**): schematic illustration of CPNs for biomedical applications. (**Right**): schematic representation of donor and acceptor molecules developed for low-band-gap conjugated polymers and photophysical processes using a Jablonski diagram. (**a**) Representative chemical structures of monomers for low-band-gap polymers. (**b**) Absorption; internal conversion; fluorescence, intersystem crossing, phosphorescence; photothermal. (**c**) Cartoon presentation of the synthesis of CPNs. (**d**) Cartoon representation of surface functionalization of CPNs. Reproduced from Ref. [13] with permission from the ACS publisher.

**Figure 2 microorganisms-11-02006-f002:**
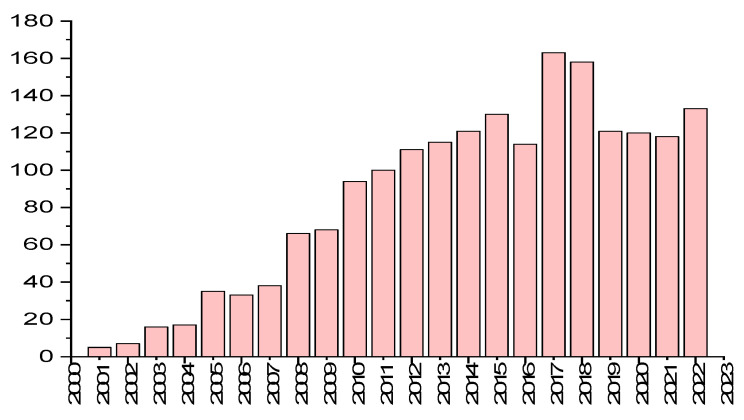
Histogram showing the number of publications in recent years according to Web of Science on the topic of “conjugated polymers” and nanoparticles.

**Figure 3 microorganisms-11-02006-f003:**
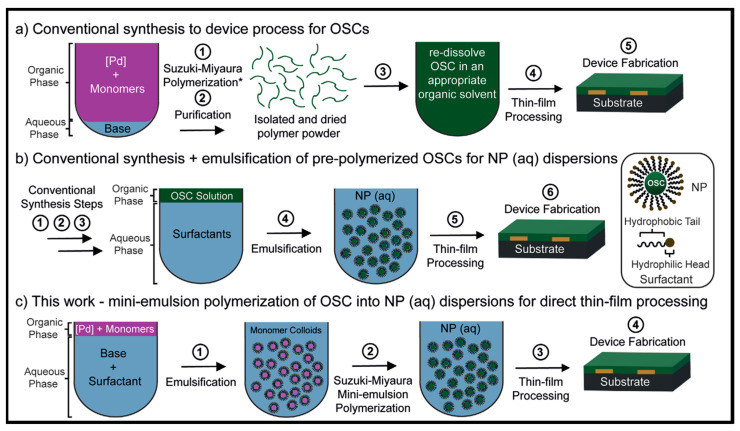
A cartoon overview of the synthesis-to-device process of OSCs. From top to bottom: (**a**) Conventional synthesis: (1) Polymerization based on organometallic cross-coupling reactions in a Ibiphasic system, (2) purification of OSCs using organic solvent (Soxhlet or preparatory size-exclusion chromatography); (3), (4) thin-film processing from halogenated solvents. (**b**) Process for the preparation of water-dispersible CPNs, (**c**) water in oil mini-emulsion Suzuki–Miyaura polymerization. Reproduced from [31]. Figure is available via license: Creative Commons CC BY license.

**Figure 4 microorganisms-11-02006-f004:**
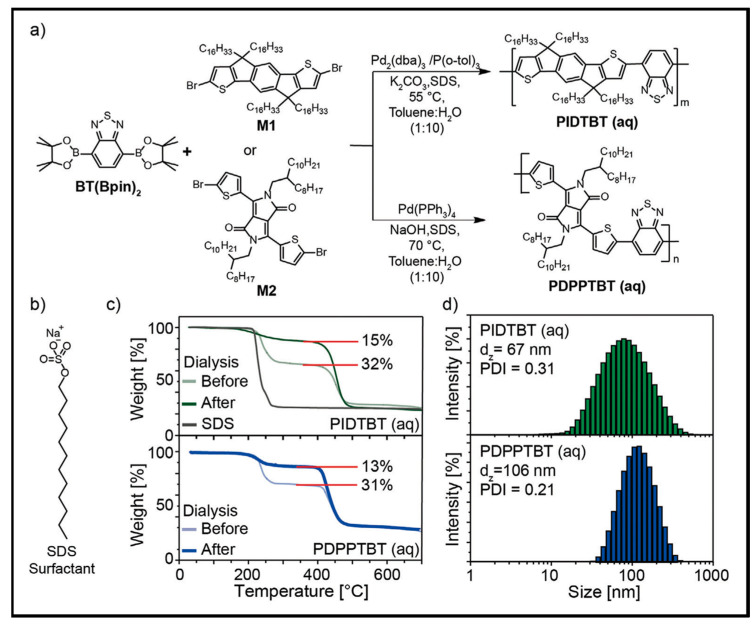
(**a**) Synthetic procedures for the copolymers: reagents and reaction conditions for mini-emulsion Suzuki–Miyaura polymerization; (**b**) Chemical structure of SDS; NP dispersion characterization; (**c**) TGA characterization of NPs showing the elimination of the excess of SDS via dialysis, and (**d**) DLS characterization of NPs dispersion. Reproduced from [31], figure available via license: Creative Commons CC BY license.

**Figure 5 microorganisms-11-02006-f005:**
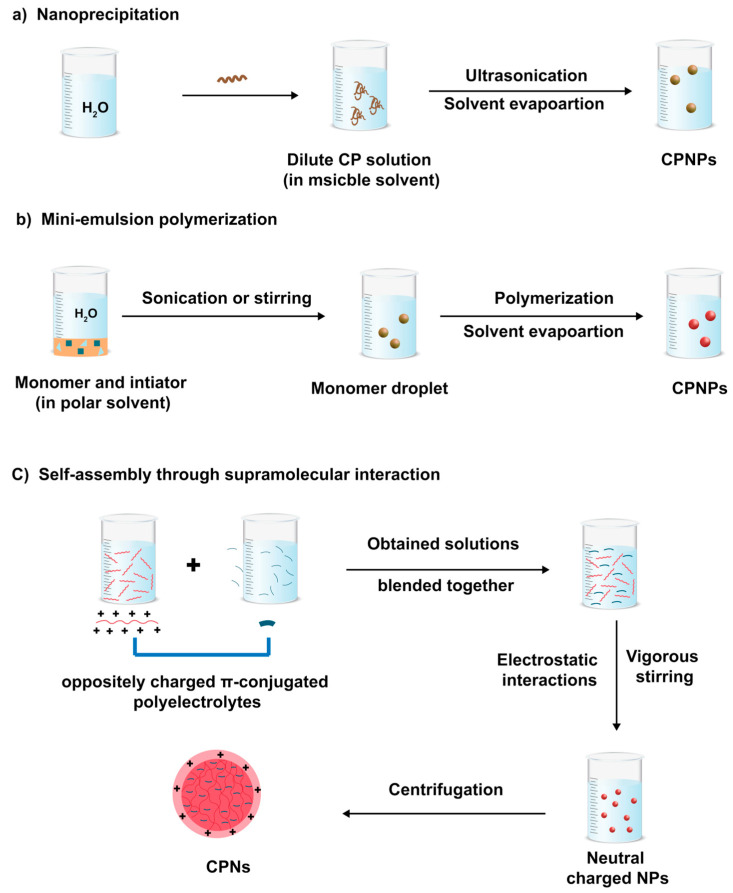
General schematic diagram for the preparation methods of conjugated polymer nanoparticles.

**Figure 6 microorganisms-11-02006-f006:**
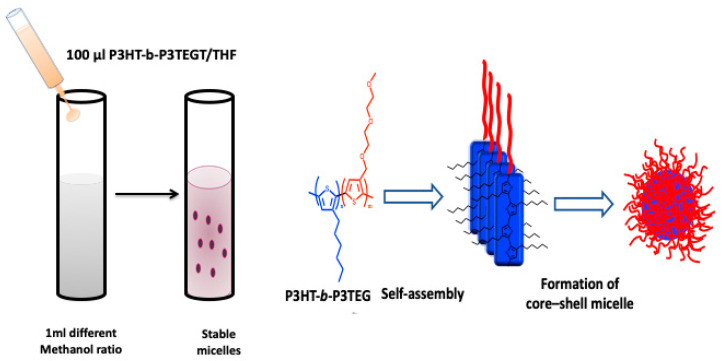
Cartoon overview of an oil-in-water mini-emulsion to prepare P3HT-*b*-P3TGT micelles. Illustration of the chemical structure of P3HT-*b*-P3TEGT and its self-assembly into nano-micelles. Reproduced from Ref. [35].

**Figure 7 microorganisms-11-02006-f007:**
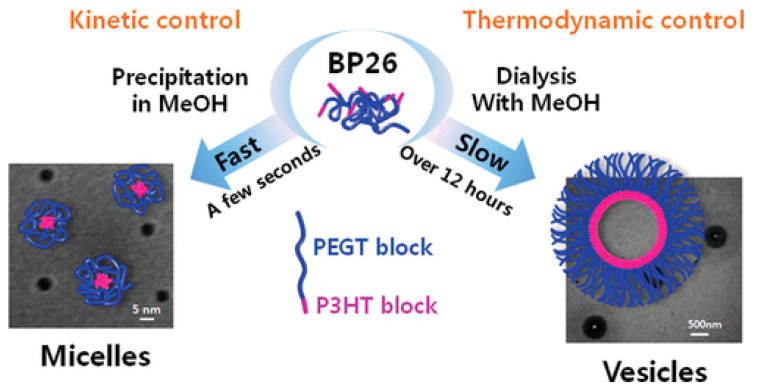
Schematic illustration of the self-assembly processes during precipitation (**left**) or dialysis (**right**) reproduced from Ref. [37] with permission from ACS.

**Figure 8 microorganisms-11-02006-f008:**
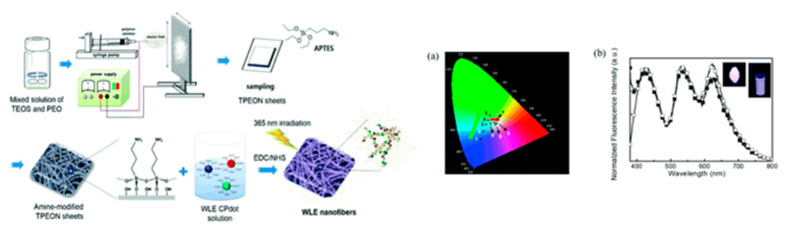
(**Left**) schematic presentation showing the fabrication of CPNs for white light emission. (**Right**) (**a**) CIE coordinates of CPN dispersion (point 8) (0.342, 0.340) and WLE nanofiber (point 9) (0.333, 0.331). The arrows and numbers are related to the composition of CPNs in solution. (**b**) Fluorescence emission of CPNs dispersion inset photographs: WLE CNPs dispersion (**right**) and WLE nanofiber (**left**). Reproduced with permission from RSC [39].

**Figure 9 microorganisms-11-02006-f009:**
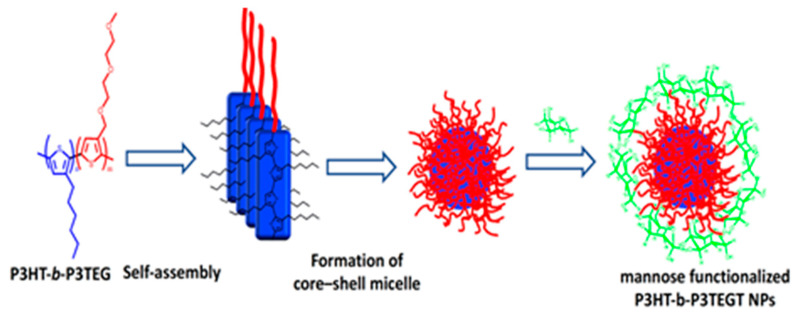
Cartoon overview of self-assembly of P3HT-b-P3TEGT in methanol. In methanol, π−π interactions help to self-assembly the nanoparticles, in which hydrophobic moieties, blue segments, come together to form the inner core and the PEG segments, in red, on the outer surface of the NPs. Mannose was coated on P3HT-b-P3TEGT nanoparticles by noncovalent interactions between hydrophilic TEGT segments on the outer surface of the nanoparticles and hydroxyl groups of the mannose.

**Figure 10 microorganisms-11-02006-f010:**
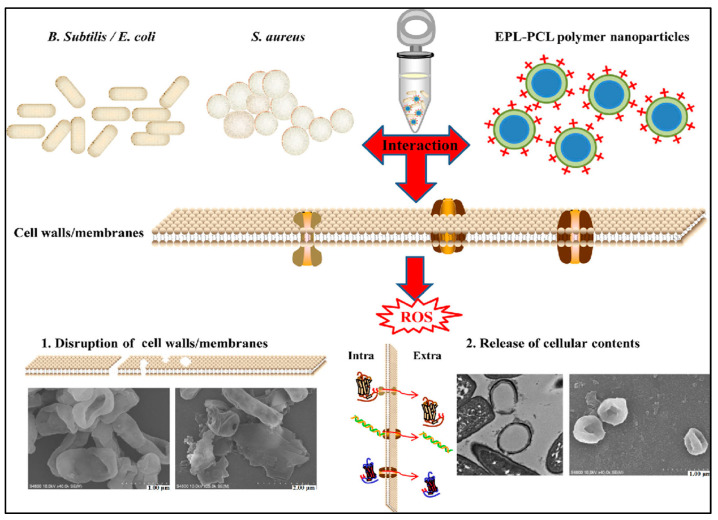
A cartoon scheme to illustrate the antibacterial mechanisms of the EPL–PCL NPs. Reproduced from Ref. [62] with permission from Elsevier. The scale bare in each image is 1 μzm.

**Figure 11 microorganisms-11-02006-f011:**
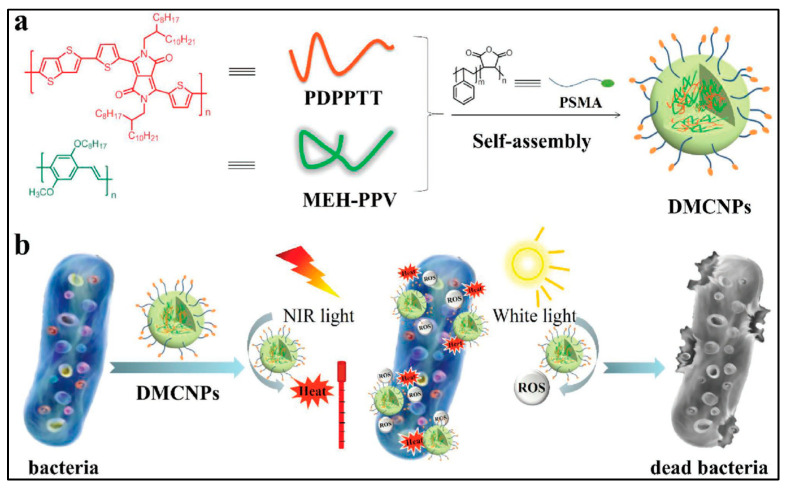
From top to bottom: (**a**) chemical structures of photodynamic conjugated polymers, (**b**) cartoon schematic illustration of photothermal and photodynamic therapy of CPNs for antimicrobial application. Reproduced from [64] with the permission of copyright Wiley.

**Figure 12 microorganisms-11-02006-f012:**
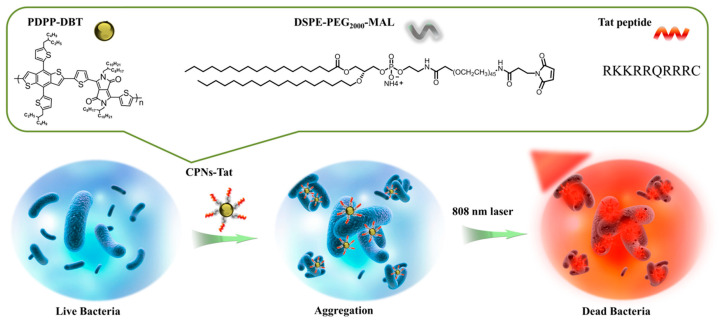
(**Top**) chemical structure of photothermal conjugated polymer and penetrating peptide. (**Bottom**) cartoon illustration of the photothermal killing of microbes by using CPNs-Tat as a photothermal agent upon NIR light. Reproduced from Ref. [64] with permission from ACS.

**Figure 13 microorganisms-11-02006-f013:**
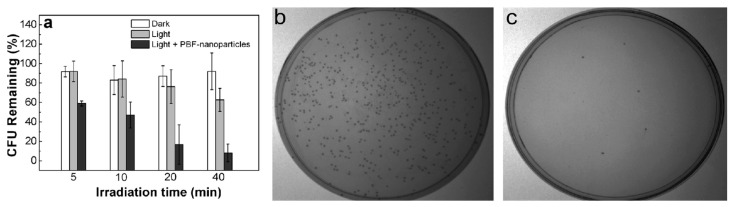
(**a**) Biocidal activity of PBF NPs toward Ampr *E. coli* in the dark and under white light for different irradiation times. (**b**) Plate photographs of Ampr *E. coli* incubated with PBF NPs for 40 min in the dark (**c**) and upon exposure to white light. Reproduced from Ref. [33] with permission from ACS.

**Figure 14 microorganisms-11-02006-f014:**
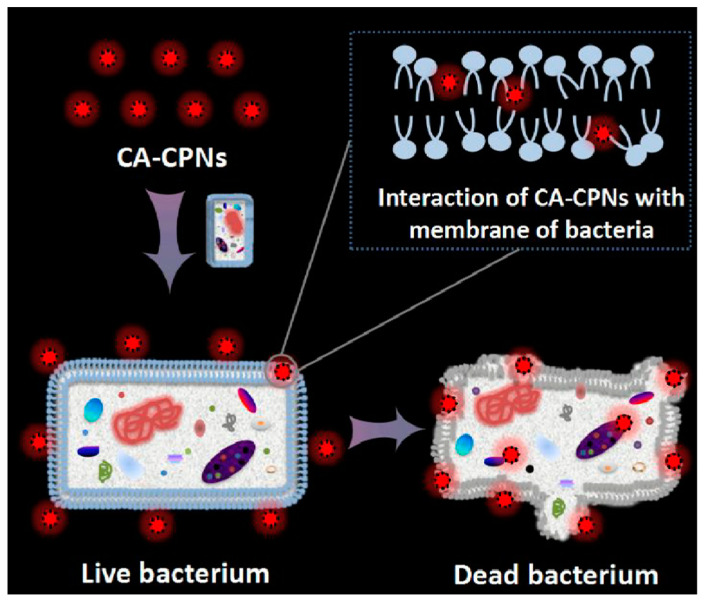
Schematic of dark antibacterial activity of cationic conjugated polymer NPs reproduced from [70] with permission from ACS.

**Figure 15 microorganisms-11-02006-f015:**
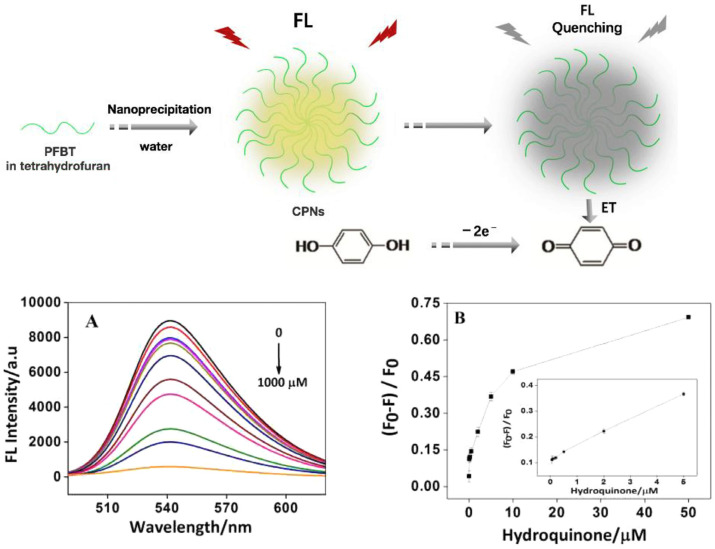
(**Top**): Schematic illustration of fabrication of the hydroquinone biosensor based on fluorescent CPNs. (**Bottom**): (**A**) Fluorescence spectra at different concentrations of hydroquinone range from 0.01 mM to 1000 mM. (**B**) Plot of fluorescence peak intensities versus the hydroquinone concentration. Inset: linear relationship between the fluorescence peak intensity and the logarithm of hydroquinone concentration. Reproduced from Ref. [79]. with permission from Elsevier.

**Figure 16 microorganisms-11-02006-f016:**
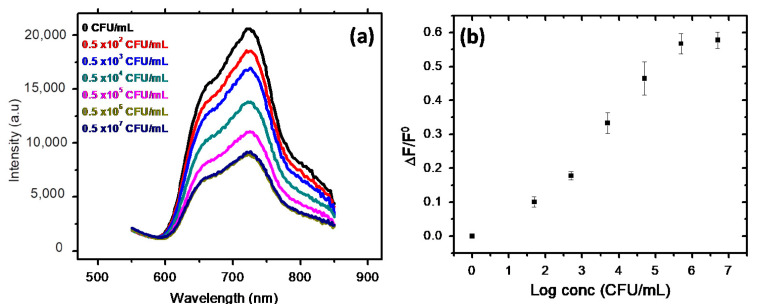
(**a**) Fluorescence spectra of CTAB-P3HT NPs at different concentrations of *E. coli* range from 0 CFU/mL to 0.5 × 10^7^ CFU/mL, (**b**) calibration curve, a plot of fluorescence peak intensities, ΔF/F^0^ vs. Log concentrations of *E. coli* loads.

**Figure 17 microorganisms-11-02006-f017:**
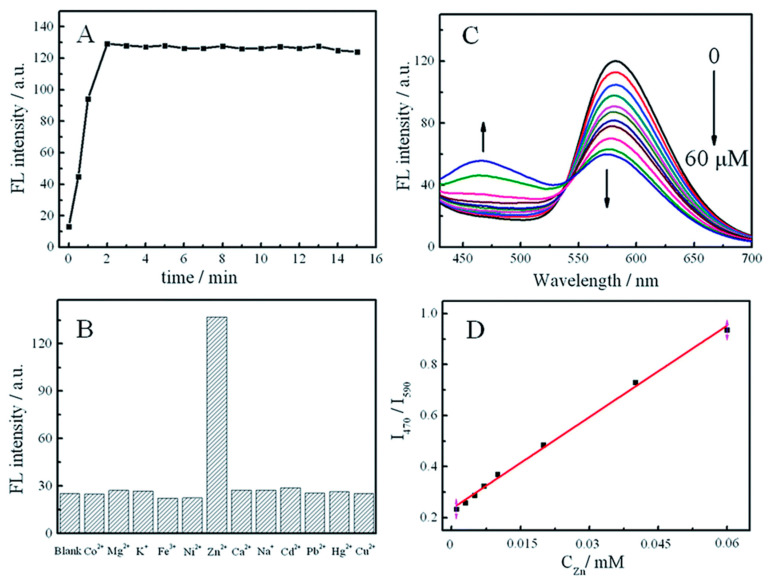
(**A**) Plot of fluorescence intensity versus time at 470 of 0.02 mg mL^−1^ Tb-HDBB-CPNs to 0.1 mM Zn^2+^ in pH 7.0. (**B**) A plot of selectivity of Tb-HDBB-CPN-based sensor for Zn^2+^ to other cations. (**C**) Fluorescence spectra of Tb-HDBB-CPNs probe upon addition of different concentrations of Zn^2+^. (**D**) The linear plot of the variation of the maximum of the emission wavelength at 470/590 versus the concentration of Zn^2+^. Reproduced from Ref. [79], copyright figures available under a Creative Commons Attribution Non-Commercial 3.0 Unported License.

**Figure 18 microorganisms-11-02006-f018:**
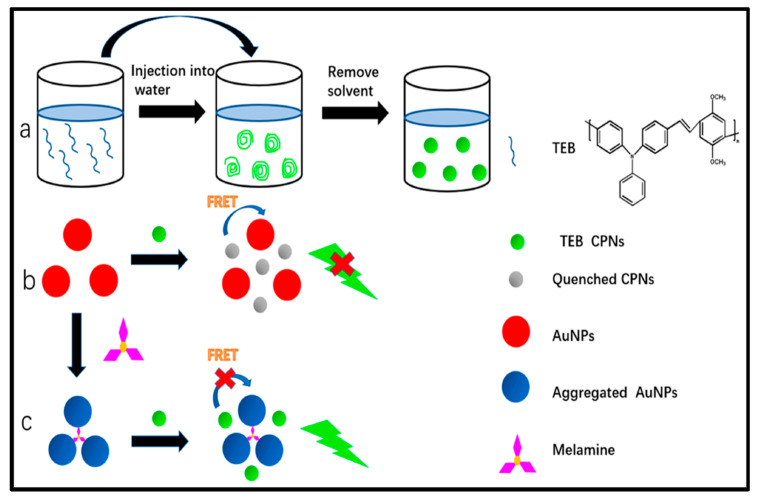
Schematic illustration of working principle of melamine FRET biosensor based on CPNs-and AuNPs; (**a**) mini-emulsion preparation process of CPNs; (**b**) CPNs fluorescence quenching by a FRET from AuNPs; (**c**) restoring of the fluorescence of CPNs after adding melamine. Reproduced from [81]. Copyright figure available under Creative Commons Attribution (CC BY) license.

**Figure 19 microorganisms-11-02006-f019:**
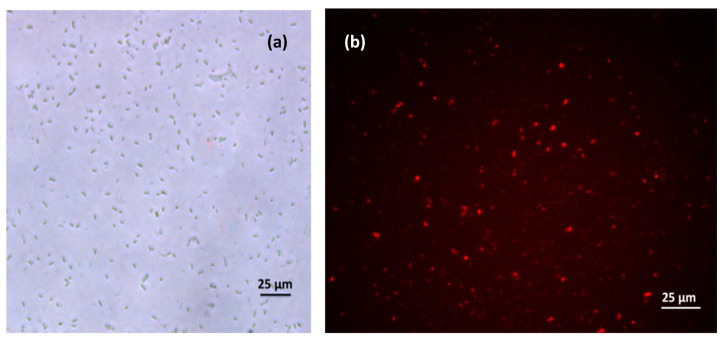
Fluorescence confocal microscopy images showing *E. coli* incubated with CTAB-P3HT NPs, (**a**) Phase-contrast image and (**b**) fluorescence image.

**Figure 20 microorganisms-11-02006-f020:**
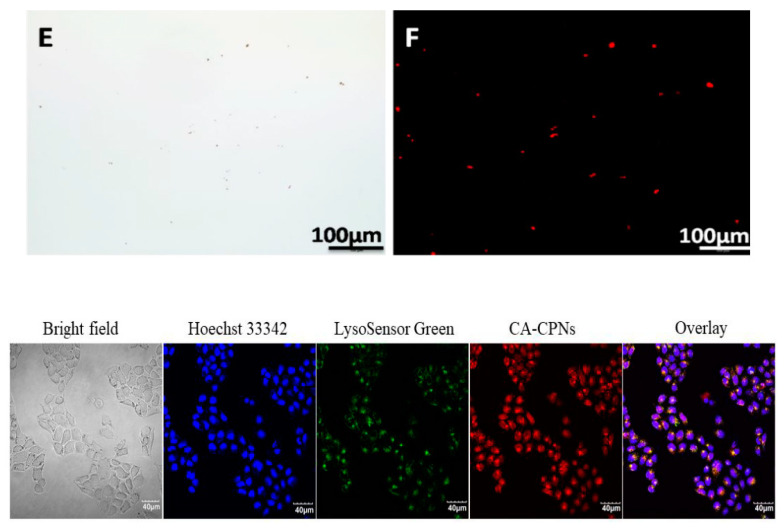
(**Top left**) Phase-contrast image and (**Top right**) fluorescence image of *E. coli* incubated with CA-CPNs. The excitation wavelength was 530 nm. [CA − CPNs] = 1.0 μg/mL. (**bottom**) Confocal fluorescence microscopy images of MCF-7 treated with 1.0 μg/mL of CA-CPNs for 18 h. Reproduced from Ref. [70] with permission from ACS.

**Figure 21 microorganisms-11-02006-f021:**
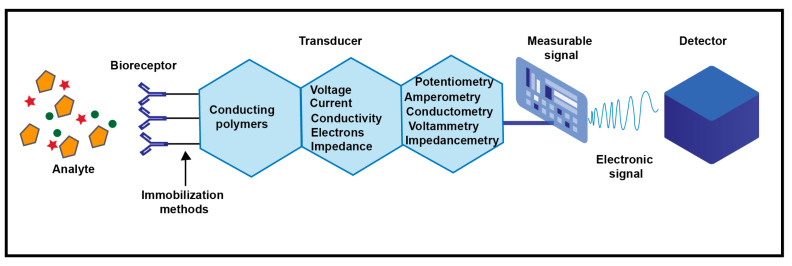
Detection principles of conducting polymer-based electrochemical biosensors.

**Table 3 microorganisms-11-02006-t003:** Recent examples of conducting polymer-based electrochemical biosensors.

Active Layer	Linear Range	Sensitivity	Detection Limit	Stability	Real Samples	Ref.
CP-based glucose amperometric biosensor						
Poly(Azure A)-Pt NP	0.02–2.3 mM	42.7μA mM^−1^ cm^−2^	7.6 μM	3 months	Fruit juice	[107]
Polycarbazole	0.01–5 mM	14.0 μA mM^−1^ cm^−2^	0.2 μM	---	---	[108]
CP-based immunosensor		Target		Detection Mode		
Polypyrrole-polythionine	0.001–100 pg/mL	Neuron-specific enolase	0.65 pg/mL	Voltammetry	---	[109]
CP-based DNA Biosensors						
Polyaniline–MoS_2_	10^−15^–10^−6^ M	DNA	10^−15^ M	Voltammetry	---	[110]
Polyaniline–graphene	10^−9^–10^−6^ M	Mycobacterium tuberculosis DNA	8 × 10^−7^ M	Voltammetry	---	[111]
CP-based Biosensors for Cancer Diagnosis		Biomarker				
Poly(triphenylamine rhodanine-3-acetic acid-co-3,4-ethoxylene dioxy thiophene)/MoS_2_/Peptide	1 × 10^−2^ ng mL^−1^	Matrix metalloproteinase-1	1 × 10^−3^ ng mL^−1^	CV ^a^	Lung cancer cells	[112]
PEDOT/peptide ^b^	50–10^6^ cells mL^−1^	CTC ^c^	17 cells mL^–1^	DPV ^d^	Breast cancer cells	[113]

Abbreviations: ^a^ cyclic voltammetry: CV; ^b^ poly(3,4-ethylenedioxythiophene): PEDOT; ^c^ circulating tumour cells: CTC; ^d^ differential pulse voltammetry: DPV.

## Data Availability

Not applicable.

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
