# Peer review of "π-Conjugated Polymer Nanoparticles from Design, Synthesis to Biomedical Applications: Sensing, Imaging, and Therapy"

_microorganisms, 2023, doi:10.3390/microorganisms11082006_

Round 1

Reviewer 1 Report

In this review, Yassar et al discussed studies on conjugated polymer nanoparticles (CPNs) and their biomedical applications. This is an interesting and important review paper. I recommend that paper should be accepted for publication with major revisions. Below is my detailed report.

1.       In page 3 section 2, the author introduced post-polymerization followed by direct polymerization. In later paragraphs, an alternative order is used where direct polymerization is mentioned before post-polymerization. Maintaining a single order of these two strategies would help the reader understand.

2.       Study mentioned in line 265 and in Figure 6 comes under the section “optical properties” but that study does not appear to mention optical properties. Further elaboration would help the reader make connections. Additionally, “OSCs” full form is not given in that paragraph or figure caption.

3.       In section 2.4 authors prepare the reader to understand multiple properties of the CNPs but later only optical properties are given as section 2.4.1. And then a new section is opened as morphological and chemical properties of CPNs in 2.5.

4.       On pages 23-24 authors mentioned about high cytotoxicity of inorganic quantum dots and they have given CPNs as alternatives with low cytotoxicity. There is no comparative data or reference material mentioned to reinforce these. They could add one or two references that compare the cytotoxicity of inorganic quantum dots and CPNs.

5.       In line 718 bioimaging is given as section 4.3 and mentions fluorescence again as previous sections did. And authors opened a new section with the same number 4.3 as electrochemical sensing. It may have been that the first 4.3 section was mistyped.

none

Author Response

Point 1

In page 3 section 2, the author introduced post-polymerization followed by direct polymerization. In later paragraphs, an alternative order is used where direct polymerization is mentioned before post-polymerization. Maintaining a single order of these two strategies would help the reader understand.

Response:

Thank you for point this out, in the revised version, we have corrected this by using some word for each approache, page 3.

Point 2 " Study mentioned in line 265 and in Figure 6 comes under the section “optical properties” but that study does not appear to mention optical properties. Further elaboration would help the reader make connections. Additionally, “OSCs” full form is not given in that paragraph or figure caption."

Response: We agree with the reviewer's comment; and we have moved this paragraph to the end of the section 2.2.1, page 6. The  new section : 2.4 Effect of synthesis conditions on the optical properties of CNPs is fully devoted to the optical properties of CPNs, page 10.

Point 3. " In section 2.4 authors prepare the reader to understand multiple properties of the CNPs but later only optical properties are given as section 2.4.1. And then a new section is opened as morphological and chemical properties of CPNs in 2.5"

Response: We agree with the reviewer's comment; and we have modified the main section and subsection, additionally, a table of contents has been added, page 1, to improve overall cohesion of the manuscript.

Point 4 " On pages 23-24 authors mentioned about high cytotoxicity of inorganic quantum dots and they have given CPNs as alternatives with low cytotoxicity. There is no comparative data or reference material mentioned to reinforce these. They could add one or two references that compare the cytotoxicity of inorganic quantum dots and CPNs."

Response: We agree that there is no comparative discussion of the nanoparticles' toxicity. We have added on paragraph on page 13, to mention that. 

It is note-worthy that compared with the inorganic quantum dots, CPNs exhibit less toxicity, which is confirmed by methyl thiazolyl diphenyl tetrazolium assay[43]. Although there is no comparative study of the toxicity of CNPs and their inorganic counterparts, the low cytotoxicity of the CNPs is due to the fact that the conjugated polymer structures do not contain any toxic elements and they are highly biocompatible.

Point 5" In line 718 bioimaging is given as section 4.3 and mentions fluorescence again as previous sections did. And authors opened a new section with the same number 4.3 as electrochemical sensing. It may have been that the first 4.3 section was mistyped."

Response: This point has been addressed by restructuring the manuscripts, modified the main section, subsection, and adding a table of contents.

Reviewer 2 Report

The paper by Elgiddawy is an interesting overview about the biomedical applications of conjugated polymer nanoparticles with large π-conjugated backbones and semiconducting properties. The topic of the paper is worthy of investigation and well fits with the scope of the journal.

This reviewer has no specific concerns about the organization of the paper and the presentation of the literature overview, and thus my recommendation of that the paper should be eventually published. Nevertheless, before publications, authors are encouraged to consider the following points:

-          Improving the conclusion section. Authors should better emphasize their own point of view and clearly indicate the future perspectives in the field

-          Please fill the last sections of the paper (authors contribution, etc.)

Author Response

Point 1 " Improving the conclusion section. Authors should better emphasize their own point of view and clearly indicate the future perspectives in the field "

Response: This observation is correct. We have changed the conclusion to "Conclusions and Perspectives" by adding new perspectives. page 38.

Point 2: " Please fill the last sections of the paper (authors contribution, etc.)"

Response: We have added the contribution of the authors.
